# LOW-RANK ADVERSARIAL PGD ATTACK

## ABSTRACT

Adversarial attacks on deep neural networks have become essential tools for studying model robustness, with Projected Gradient Descent (PGD) being widely adopted due to its effectiveness and computational efficiency. In this work, we provide empirical evidence that PGD perturbations are, on average, low-rank, with their magnitude concentrated in the bottom part of the singular value spectrum across CIFAR-10 and ImageNet datasets and multiple architectures. Building on this insight, we introduce LoRa-PGD, a simple low-rank variation of PGD that directly computes adversarial attacks with controllable rank. Through extensive experiments on different datasets and models from the RobustBench ModelZoo, we demonstrate that LoRa-PGD systematically outperforms or matches standard PGD in terms of robust accuracy and achieves performance comparable to AutoAttack while requiring orders of magnitude less computational time. Additionally, we show that models adversarially trained with LoRa-PGD are consistently more accurate and more robust against full-rank attacks compared to standard adversarial training, suggesting that low-rank perturbations capture important but otherwise hidden vulnerability patterns.

## 1 MOTIVATION AND CONTRIBUTIONS

Adversarial attacks, characterized by subtle data perturbations that destabilize neural network predictions, have been a topic of significant interest for over a decade Szegedy et al. (2013); Goodfellow et al. (2014); Moosavi-Dezfooli et al. (2016); Carlini & Wagner (2017). These attacks have evolved into various forms, depending on the knowledge of the model's architecture (white-box, gray-box, black-box) Vivek et al. (2018), the type of data being targeted (graphs, images, text, etc.) Entezari et al. (2020); Sun et al. (2022); Goodfellow et al. (2014); Zhang et al. (2020), and the specific adversarial objectives (targeted, untargeted, defense-oriented) Yuan et al. (2019); Madry et al. (2017) and parameter free Croce & Hein (2020).

While numerous defense strategies aim to broadly stabilize models against adversarial attacks, independent of the specific attack mechanism Cisse et al. (2017); Galloway et al. (2018); Ghiasi et al. (2024); Savostianova et al. (2024), the most effective and widely-used defenses focus on adversarial training, where the model is trained to withstand particular attacks Madry et al. (2017); Wang et al. (2019). Adversarial training is known for producing robust models efficiently, but its effectiveness hinges on the availability of adversarial attacks that are both potent in degrading model accuracy and efficient in terms of computational resources. However, the most aggressive attacks often require significant computational resources, making them less practical for adversarial training. The projected gradient descent (PGD) attack Madry et al. (2017) is popular in adversarial training due to its balance between aggressiveness and computational efficiency.

Recent work has provided evidence that adversarial vulnerability may have a spectral origin. In Jere et al. (2020), the authors investigate the dependency of robustness on the rank of images in a classification dataset, with results suggesting that adversarial vulnerability may be caused by the classifier's dependency on small singular values of the input image. Similarly, Harder et al. (2021) shows that adversarially perturbed images can be identified through a discriminator neural network operating in the Fourier domain. These findings suggest that imperceptible adversarial examples may result from the network's dependency on "spectrally negligible" features of the input. This perspective also suggests that norms measuring spectral discrepancies, such as the nuclear norm, rather than norms focused on pixel-wise differences, such as the $L^2$ norm, may provide different insight into the nature of adversarial examples on images.

Building on this line of work, in this paper, we observe that perturbations generated by PGD predominantly affect the lower part of the singular value spectrum of input images, indicating that these perturbations are indeed approximately low-rank. Additionally, we find that the magnitude of PGD-generated attacks differs significantly between standard and adversarially trained models when measured by their nuclear norm, which sums the singular values of the attack. This metric provides insight into the frequency profile of the attack when analyzed using the singular value decomposition and aligns with known frequency profiles observed under discrete Fourier and discrete cosine transforms of PGD attacks Yin et al. (2019); Maiya et al. (2021).

Following these observations, we introduce LoRa-PGD, a simple yet effective low-rank variation of PGD designed to compute adversarial attacks with controllable rank. Our results demonstrate that perturbing only a small percentage of the image's singular value spectrum can achieve accuracy degradation comparable to full-rank PGD when the perturbation size is measured using the pixel-wise $L^2$ norm, or superior performance when measured with the nuclear norm, while requiring smaller run time. Moreover, when used to adversarially train a model from scratch, the computed low-rank perturbations prove more effective, resulting in models that are significantly more resilient to adversarial attacks.

## 2 RELATED WORK

Following the seminal contributions of Szegedy et al. (2013); Goodfellow et al. (2014), the past decade has witnessed significant efforts to understand the stability properties of neural networks. This has led to the development of various adversarial attacks, categorized broadly into model-based approaches Poursaeed et al. (2018); Laidlaw & Feizi (2019); Xiao et al. (2018); Song et al. (2018) and optimization-based methods Moosavi-Dezfooli et al. (2016); Goodfellow et al. (2014); Madry et al. (2019). Concurrently, several robustification strategies have been proposed to mitigate these attacks Madry et al. (2017); Bai et al. (2021). A key challenge in these approaches is the dependency on the availability of adversarial examples during the training phase, making the efficiency of generating such examples a central concern. Furthermore, the perceptibility of adversarial attacks is another critical factor that has been explored in various studies Croce & Hein (2019); Qin et al. (2019).

**Adversarial Attacks and Training**  The discovery of adversarial examples has catalyzed extensive research on developing robust models. Typically, ensuring robustness against adversarial attacks is computationally intensive, as it often requires the generation of adversarial examples during training Goodfellow et al. (2014); Xiao et al. (2018). However, adversarial training is not the only method for achieving robustness. Other approaches involve imposing constraints on models to enhance their robustness Leino et al. (2021); Savostianova et al. (2024); Zhang et al. (2022); Fazlyab et al. (2023). The primary distinction between these approaches lies in their trade-offs: adversarial training techniques generally offer higher robustness but are more expensive due to the need for real-time adversarial example generation during training. In contrast, approaches aiming at improving intrinsic model robustness may be less efficient in terms of robustness accuracy but are typically less expensive and more general, in the sense that they offer robustness properties that do not depend on the training data or the specific attack, and thus are more consistent across different scenarios.

**Spectral and Frequency Properties of Adversarial Attacks**  Recent research has focused on the spectral and frequency properties of adversarial attacks, revealing key insights into their nature. Jere et al. (2020) investigate the relationship between adversarial robustness and the rank properties of images, demonstrating that adversarial vulnerability may stem from classifiers' dependency on small singular values of input images. Similarly, Harder et al. (2021) show that adversarially perturbed images can be detected through spectral analysis in the Fourier domain, further supporting the connection between adversarial examples and spectrally negligible features. Additional studies have explored frequency-domain characteristics of adversarial attacks Maiya et al. (2021); Luo et al. (2022); Guo et al. (2020); Sharma et al. (2019); Yin et al. (2019), revealing that adversarial perturbations often concentrate energy in specific frequency bands.

**Low-Rank Structures**  Low-rank structures are widely explored and used in deep learning and scientific computing to develop more effective and memory-efficient algorithms. Classical applications range from singular value decomposition for image compression to tensor networks used for effi-

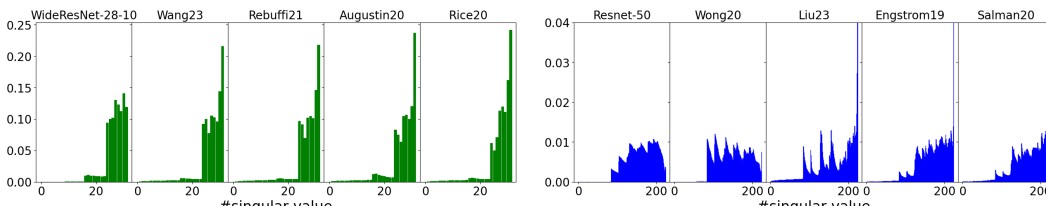

Figure 1: Relative magnitude of singular value change for PGD-attacked images, demonstrating the low-rank nature of adversarial perturbations. The plots show the average relative change in singular value magnitudes across the entire spectrum for 5000 images from CIFAR-10 (left, green) and ImageNet (right, blue) datasets and across ten different neural network models. The x-axis represents the singular value index (ordered from largest to smallest), while the y-axis shows the relative change in magnitude. Experimental details are described in Section 4.

ciently representing quantum wave functions, as well as neural network layers Chen (2018); Román (2019); Ceruti et al. (2021); Pan et al. (2019). Low-rank methods are effectively used to develop recommender systems Koren et al. (2009), latent factor models, and principal component analysis Jolliffe & Cadima (2016); Yang (2021). Low-rank factorizations have also proven very effective in deep learning, for their ability to provide efficient neural network post-training compression Wang et al. (2020); Hsu et al. (2022); Yuan et al. (2023); Pan et al. (2019), compression during training Lialin et al. (2024); Zhao et al. (2024); Schotthöfer et al. (2022); Idelbayev & Carreira-Perpinan (2020); Novikov et al. (2015), and efficient fine-tuning with low-rank adapters Hu et al. (2022); Zhang et al. (2023); Schotthöfer et al. (2025); Mao et al. (2024).

Despite the extensive application of low-rank methods in neural network optimization and compression, their direct application to adversarial attacks remains underexplored. In this work we address this gap by introducing LoRa-PGD, a simple and memory and time-efficient attack strategy superimposing a low-rank structure in the perturbations.

## 3 LOW-RANK PGD-STYLE ATTACK

### 3.1 THE SINGULAR SPECTRUM OF PGD ATTACKS

Let us consider a general setup based on the image classification task. Let $X \in \mathcal{C} \subseteq \mathbb{R}^{C \times N \times M}$ be an input data tensor with $C$-channels ($C = 3$ for RGB images) of sizes $N \times M$ and $Y \in \mathbb{R}^D$ the target tensor. Assume also we have a model $f_\theta : \mathbb{R}^{C \times N \times M} \to \mathbb{R}^D$ that has been trained with respect to a loss function $\ell : \mathbb{R}^D \times \mathbb{R}^D \to \mathbb{R}$. An untargeted adversarial attack on the input data $X$ can be naturally formulated as a perturbation $\delta X^*$ of bounded norm corresponding to the largest induced loss change for the model output, or, in other words, a solution to the constrained optimization problem:

$$
\begin{cases}
\delta X^* \in \underset{\delta X \in \mathbb{R}^{C \times N \times M}}{\arg\max} \ \ell(f_\theta(X + \delta X), Y) \\
\text{s.t.} \quad \|\delta X\|_\omega \leq \tau, \ X + \delta X \in \mathcal{C}
\end{cases}, \tag{1}
$$

where $\|\cdot\|_\omega$ is a generic norm applied channel-wise (see Definition A.2) and $\tau > 0$ is the perturbation norm or perturbation budget. The latter is used to control the size of the attack, and therefore it can be interpreted as a measure of the perceivability of the adversarial perturbation. Note that, in practical applications and for general neural networks, the optimization problem formulated in Equation (1) is non-convex but the constraint is compact, guaranteeing the existence of at least one solution.

The choice of the norm $\|\cdot\|_\omega$ influences the nature of the attack $\delta X^*$, with common choices being pixel-wise $L^p$ norms such as $\|\cdot\|_\infty$ (entry-wise maximum) and $\|\cdot\|_{L^2}$ (entry-wise Euclidean norm). Importantly, the norm $\|\cdot\|_\omega$ may directly affect both the effectiveness and the perceptibility of the attack, see e.g. Beerens & Higham (2023); Sharif et al. (2018). Even though all norms are equivalent in finite-dimensional vector spaces, they measure different aspects of the perturbation and an attack that is small under one norm may still be large when measured with another norm and vice-versa.

Although obtaining a global optimizer $\delta X^*$ for adversarial attacks is challenging due to the problem's inherent complexity, effective approximations can be efficiently derived. One of the most widely

adopted and computationally efficient methods is the Projected Gradient Descent (PGD) scheme Madry et al. (2017). This method approaches the problem (1) by iteratively taking steps in the direction of the (positive) gradient of the loss projected onto the unit $\|\cdot\|_\omega$-sphere.

Traditionally, PGD is applied to the flattened image tensor $X$, treated as a vector of pixels and channels. However, this flattening process ignores the spectral structure of the image and thus overlooks potentially important information in the attack. By reshaping the PGD attack back into a tensor and analyzing the singular values of its channels, we observed an intriguing low-rank pattern. Specifically, we observed that the PGD attack frequently influences only a small subset of the singular values of the original image, implying that the perturbations produced by PGD are numerically of low rank. Figure 1 illustrates this phenomenon: the figure compares the singular values of 5000 images before and after a targeted attack computed using PGD, by showing the average relative change in the magnitude of the singular values across the entire singular value spectrum. The images are collected from the CIFAR-10 and ImageNet datasets and their attacks are computed using ten different standard neural network models, later described in Section 4.

Building on this observation, in Section 3.2, we propose a variation of PGD attack which exploits this implicit bias by directly superimposing a low-rank structure on the perturbations. This allows more efficient attacks that require less memory allocation and are more effective than classic PGD for a given compute time budget. This improved adversarial performance is also reflected in a corresponding adversarial training robustness. When used to adversarially train a model from scratch, the computed low-rank perturbations are much more effective, resulting in models that are significantly more resilient to adversarial attacks.

## 3.2 LoRa-PGD: Low-rank PGD attack

Similar to what is done in the context of parameter-efficient adaptation Hu et al. (2022), we consider here a variation of PGD that directly searches for a low-rank structured attack.

Recall that a PGD attack is a way to produce a fast approximate solution to (1). Precisely, for $L^2$ based attacks, the final attack is produced as the limit point of a projected gradient ascent sequence

$$\delta X_{k+1} = \Pi\Big(\delta X_k + \tau \frac{\nabla_{\delta X}\ell(f_\theta(X + \delta X_k), Y)}{\|\nabla_{\delta X}\ell(f_\theta(X + \delta X_k), Y)\|_{L^2}}\Big), \tag{2}$$

where $\Pi$ is the projection onto the constraint set $\mathcal{C}$. A similar formula can be obtained for the gradient ascent direction computed on a generic $L^p$ norm, as we recall in Appendix B. In typical image applications, $\mathcal{C}$ describes the maximal and minimal values each RGB channel can take (thus, $\mathcal{C}$ is an hypercube) and the projection with respect to the $L^2$ metric can be computed in closed form, and it is given by the $\Pi = \mathrm{clamp}$ entry-wise function. The final PGD attack approximates $\delta X^*$ and is used as the candidate maximizer of the loss function on the constraint $\mathcal{C}$.

A low-rank attack can be formulated by looking at a rank-constrained version of (1):

$$\begin{cases} \delta X^* \in \underset{\delta X \in \mathbb{R}^{C \times N \times M}}{\arg\max} \ \ell(f_\theta(X + \delta X), Y) \\ \|\delta X\|_\omega \leq \tau, X + \delta X \in \mathcal{C}, \mathrm{rank}(\delta X) \leq r \end{cases} \tag{3}$$

where $\mathrm{rank}$ is intended channel-wise, as detailed in Definition A.2. Given the complex geometrical nature of the rank constraint, in the same spirit of Hu et al. (2022), one simple way to ensure it is to superimpose a low-rank representation for the perturbation, i.e. find the solution of the following optimization problem:

$$\begin{cases} (\delta U^*, \delta V^*) \in \underset{\substack{\delta U \in \mathbb{R}^{C \times N \times r}, \\ \delta V \in \mathbb{R}^{C \times r \times M}}}{\arg\max} \ \ell(f_\theta(X + \delta U \otimes_C \delta V), Y) \\ \|\delta U \otimes_C \delta V\|_\omega \leq \tau, \ X + \delta U \otimes_C \delta V \in \mathcal{C} \end{cases} \tag{4}$$

with $\otimes_C$ denoting the channel-wise tensor multiplication where each channel is the external product of the corresponding modes in $\delta U$ and $\delta V$. See Definition A.2 for a precise formulation. Given the large dimension of the optimization problem for typical deep-learning models, an approximation $(\delta U^*, \delta V^*)$ can be obtained in a PGD-like fashion by computing several steps of gradient ascent with respect to the smaller variables $\delta U$ and $\delta V$.

---

**Algorithm 1** Pseudocode of (**LoRa-PGD**): low-rank gradient attack

---

**Require:** $X \in \mathbb{R}^{B \times C \times N \times M}$ input tensor; $Y$ target vector; $r$ perturbation rank; $\tau$ perturbation size.
  $\delta U \leftarrow \text{normalize}_\omega(\text{init}([B, C, N, r]))$
  $\delta V \leftarrow \text{normalize}_\omega(\text{init}([B, C, r, M]))$
  **for** step in steps **do**
   $\widetilde{X} = \text{clamp}(X + \tau \cdot \text{normalize}_\omega(\delta U \otimes_C \delta V), 0, 1)$
   $\delta U \mathrel{+}= \nabla^\omega_{\delta U} \ell(f_\theta(\widetilde{X}), Y)$
   $\delta V \mathrel{+}= \nabla^\omega_{\delta V} \ell(f_\theta(\widetilde{X}), Y)$
  **end for**
  $\delta = \tau \cdot \text{normalize}_\omega(\delta U \otimes_C \delta V)$
  $\widetilde{X} = \text{clamp}(X + \delta, 0, 1)$

---

In Algorithm 1, we present the pseudocode of the resulting LoRa-PGD $\|\cdot\|_\omega$-attack. We specify that $\nabla^\omega$ refers to the gradient with respect to the norm $\|\cdot\|_\omega$ as in Definition A.3 and that the clamp function is the one defined in `torch`, which is performed entry-wise. Moreover, we specify that the normalize$_\omega$ function is just division by the norm, i.e. $\text{normalize}_\omega(z) = z/\|z\|_\omega$.

The scheme proposed in Algorithm 1 is a low-rank variation of the standard PGD. In fact, using the update steps for $\delta U$ and $\delta V$ in Algorithm 1, we obtain the analogous of a simple PGD iteration:

$$\delta U_{k+1} = \delta U_k + \nabla^\omega_{\delta U} \ell, \quad \delta V_{k+1} = \delta V_k + \nabla^\omega_{\delta V} \ell \tag{5}$$

where the gradient is evaluated on the new image, perturbed as follows

$$X = \text{clamp}\Big(X + \tau \, \text{normalize}_\omega(\delta U_{k+1} \otimes_C \delta V_{k+1})\Big).$$

Similar to PGD iteration, clamp is the closed-form projection on the data constraint $\mathcal{C} \setminus X$. In particular, note that the gradient scheme (5) remains computationally close to a basic PGD computation since gradients $\nabla_{\delta U} \ell$ and $\nabla_{\delta V} \ell$ are one tensor multiplication away from the computation of the PGD-gradient $\nabla_{\delta X} \ell$.

Note that an alternative approach to compute low-rank attacks is to perform standard PGD and then project the resulting perturbation onto the set of rank $r$ matrices at each step (or after a certain number of steps). This approach is viable but not efficient. Projecting onto the set of rank $r$ matrices requires computing a truncated SVD, which is expensive and not well-integrated into standard autodifferentiation libraries. We do not report a numerical comparison with this approach as in our experiments, the performance obtained by this rank-projected PGD approach was the same as that of Algorithm 1, while being almost one order of magnitude slower.

**Nuclear Norm and Low-Rank Matrices** The perturbation budget $\tau$ is typically measured in terms of vector $L^p$ norms, enforcing an upper bound on the size of the perturbation pixel-wise. However, in order to measure the size of the perturbation from the spectral point of view, we additionally consider here the nuclear norm, which sums the singular values of the perturbation. Precisely, if we let $R = \min(N, M)$, then we set

$$\|X\|_{S^1} = \frac{1}{C} \sum_i^C \sum_j^R \sigma_j(X_{i,:,:}) = \frac{1}{C} \sum_i^C \|X_{i,:,:}\|_{S^1}$$

where $\sigma_j(A)$ denotes the $j$th largest singular value of $A$. Notice that this definition coincides (up to a constant factor) with the $L^1$ norm of the singular value vector, which is known to induce sparsity in the optimization problem. In particular, it represents a convex relaxation of the $L^0$ Shatten norm, and its usage is analogous to the use of the $L^1$ norm as a convex surrogate for the $L^0$ norm in compressed-sensing problems or robust PCA Simon & Holger (2013); Candes et al. (2009).

## 4 EXPERIMENTS

In order to provide a reproducible comparison, RobustBench Croce et al. (2021) was used for pretrained models, `adversarial-attacks-pytorch` Kim (2020) library was used as a basis for

Table 1: Comparison of robust accuracy $\rho$ across different datasets, models and algorithms for 50 optimization steps. For each section (each $\tau$), the best result per column is given in bold, and the second best is given in gray. AutoAttack is always the best performing one, but it is considerably slower and is not accounted for when highlighting.

| | | CIFAR-10 | | | | | ImageNet | | | | | Ranks $r = \cdot\%R$ |
|---|---|---|---|---|---|---|---|---|---|---|---|---|
| | | St. | Wa23 | Re21 | Au20 | Ri20 | St. | Wo20 | Li23 | En19 | Sa20 | |
| **LoRa-PGD** $\|\cdot\|_2 = 0.2$ | | 0.315 | 0.928 | 0.885 | 0.91 | 0.838 | 0.098 | 0.737 | 0.451 | 0.562 | 0.556 | 10% |
| | | 0.202 | 0.924 | 0.883 | 0.904 | 0.832 | 0.052 | 0.735 | 0.443 | 0.559 | 0.55 | 20% |
| | | 0.15 | 0.922 | 0.88 | 0.901 | 0.828 | 0.04 | 0.735 | 0.441 | 0.557 | 0.549 | 30% |
| | | 0.134 | 0.922 | 0.88 | **0.9** | 0.826 | 0.035 | 0.734 | 0.44 | 0.557 | 0.548 | 40% |
| | | **0.126** | 0.922 | 0.88 | **0.9** | 0.826 | **0.032** | 0.734 | 0.439 | 0.557 | 0.547 | 50% |
| Classic PGD | | 0.163 | **0.921** | **0.879** | **0.9** | **0.825** | 0.048 | **0.733** | **0.436** | **0.556** | **0.546** | 100% |
| Autoattack | | 0.092 | 0.919 | 0.876 | 0.896 | 0.821 | - | - | - | - | - | 100% |
| **LoRa-PGD** $\|\cdot\|_2 = 0.3$ | | 0.134 | 0.913 | 0.871 | 0.889 | 0.812 | 0.029 | 0.718 | 0.404 | 0.521 | 0.506 | 10% |
| | | 0.062 | 0.905 | 0.864 | 0.878 | 0.798 | 0.011 | 0.715 | 0.393 | 0.515 | 0.5 | 20% |
| | | 0.039 | 0.901 | 0.859 | 0.874 | 0.791 | 0.007 | 0.714 | 0.39 | 0.513 | 0.497 | 30% |
| | | 0.031 | **0.9** | 0.858 | 0.872 | 0.79 | **0.006** | 0.714 | 0.387 | **0.51** | 0.497 | 40% |
| | | **0.027** | **0.9** | 0.858 | 0.872 | **0.789** | **0.006** | 0.714 | 0.387 | 0.511 | **0.496** | 50% |
| Classic PGD | | 0.051 | **0.9** | **0.855** | **0.871** | **0.789** | 0.012 | **0.712** | **0.386** | 0.511 | 0.498 | 100% |
| Autoattack | | 0.015 | 0.896 | 0.848 | 0.861 | 0.78 | - | - | - | - | - | 100% |
| **LoRa-PGD** $\|\cdot\|_2 = 0.4$ | | 0.052 | 0.897 | 0.851 | 0.862 | 0.777 | 0.008 | 0.7 | 0.359 | 0.474 | 0.453 | 10% |
| | | 0.019 | 0.888 | 0.839 | 0.849 | 0.755 | 0.003 | 0.697 | 0.346 | 0.465 | 0.444 | 20% |
| | | 0.01 | 0.882 | 0.831 | 0.844 | 0.746 | 0.003 | 0.696 | 0.34 | 0.463 | **0.441** | 30% |
| | | 0.007 | 0.88 | 0.829 | **0.841** | 0.744 | **0.002** | 0.696 | 0.338 | **0.461** | 0.443 | 40% |
| | | **0.006** | 0.879 | 0.829 | **0.841** | 0.743 | **0.002** | 0.696 | **0.337** | **0.461** | 0.442 | 50% |
| Classic PGD | | 0.018 | **0.878** | **0.827** | 0.842 | **0.74** | 0.004 | **0.695** | 0.343 | 0.467 | 0.448 | 100% |
| Autoattack | | 0.002 | 0.87 | 0.817 | 0.83 | 0.733 | - | - | - | - | - | 100% |

the attack comparison, while the MAIR framework Kim et al. (2024) was used for adversarial training experiments. Implementation of the experiments is included in the supplementary material.

**Datasets** We use standard benchmark datasets for attacks on image classifiers. Specifically, CIFAR-10 Krizhevsky et al. (2009) and ImageNet Deng et al. (2009). The test sets contain 5000 examples in both cases. The former dataset has 10 classes and the latter 1000 classes. The original resolution of the images in the two datasets is substantially different: CIFAR-10 contains images whose size is $3 \times 32 \times 32$, while ImageNet contains images of size $3 \times 256 \times 256$. In practice, as a result of preprocessing ImageNet, the size is decreased to $3 \times 224 \times 224$ for all models we used.

**Models** We tested LoRa-PGD both on different standard (non-robust) models and several adversarially robust models from Robustbench ModelZoo Croce et al. (2021). As non-robust models, we used WideResNet-28-10 for all CIFAR-10 experiments, and Resnet-50 for all the ImageNet ones. As robust-models, we used Wang2023 Wang et al. (2023), Rebuffi2021 Rebuffi et al. (2021), Augustin2020 Augustin et al. (2020), Rice2020 Rice et al. (2020) for all CIFAR-10 experiments, and Liu2023 Liu et al. (2023), Salman2020 Salman et al. (2020), Wong2020 Wong et al. (2020), Engstrom2019 Engstrom et al. (2019) for all the ImageNet ones.

Methods are compared in terms of robust accuracy $\rho$, which we recall in Definition A.1. This metric represents the percentage of dataset images for which the adversarial perturbation leaves unchanged the model's output class.

**Baselines.** In the experimental section, we compare LoRa-PGD with the standard implementation of the PGD attack Madry et al. (2017) and with AutoAttack Croce & Hein (2020).

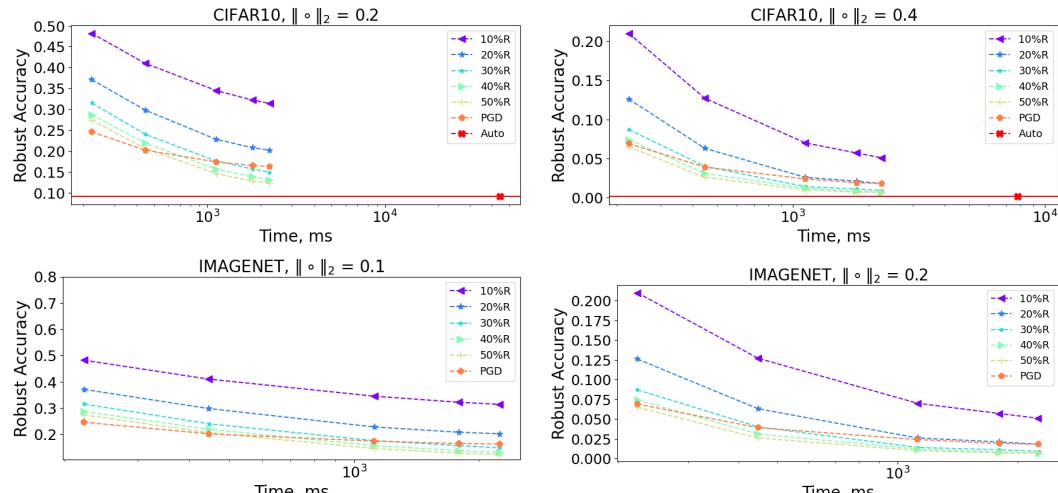

Figure 2: Comparison of different attacks' time vs robust accuracy for the Standard model on CIFAR-10 (top row) and ImageNet (bottom row).

## 4.1 RESULTS ON THE EFFECTIVENESS AND EFFICIENCY OF THE ATTACK

In Table 1 and Figure 2, we summarize the adversarial performance of LoRa-PGD against PGD and AutoAttack. For LoRa-PGD, we tested relative ranks ranging from $10\%$ to $50\%$ of the maximum rank.

In Table 1 we present results for 3 different choices of $L^2$ norm of the perturbation, for each we provide a comparison of robust accuracy for LoRa-PGD with different rank choices and using 50 steps, PGD, also with 50 steps, and AutoAttack on both CIFAR-10 and ImageNet and the model choices described above. Note that AutoAttack results for ImageNet are not provided due to time and memory constraints being exceeded by this method. We note that our method consistently outperforms PGD for the standard models, while for robust models, it often outperforms standard PGD on ImageNet and shows almost equal results for CIFAR-10. While AutoAttack outperforms both LoRa-PGD and Classic PGD in terms of robust accuracy, it requires significantly more time to compute the attack, as we discuss next.

**Attacks' Efficiency vs Run Time.**    In Figure 2, we compare the performance of each attack against computational time. We run PGD and LoRa-PGD for different steps $(5, \ldots, 50)$ as well as AutoAttack. For each of these implementations, we compute the average run time over 100 images. For both CIFAR-10 and ImageNet, LoRa-PGD with 30%-50% rank size achieves a stronger robust accuracy performance as compared to PGD at equal run time budget. While AutoAttack achieves better robust accuracy, it requires up to almost two orders of magnitude more execution time. For additional figures with wider range of $\tau$, please refer to Supplementary material Figure 5-6

**Nuclear magnitude of adversarial attacks**    Entrywise $L^p$ norms are the typical choice for measuring adversarial perturbations. However, our observations in Figure 1 suggest that PGD naturally concentrates the strength of the attack on specific portions of the singular spectrum, which could therefore be more effective. To investigate the impact that singular values have on the effectiveness of an attack, we measure the size of PGD attacks on both standard and robust models in terms of their nuclear norm. Figure 3 shows that, from the perspective of singular values, PGD generates larger attacks on standard models, while the nuclear norm of the computed attacks is smaller when applied to adversarially robust models. This observation aligns with recent analyses of the frequency profiles of PGD attacks measured under Fourier and cosine transforms (Yin et al., 2019; Maiya et al., 2021).

In order to assess the capacity of LoRa-PGD to effectively concentrate the attack on relevant portions of the singular spectrum, in Table 2 we compare PGD and LoRa-PGD, while allocating to each of the perturbations the same nuclear norm budget. The results shown there demonstrate that LoRa-PGD

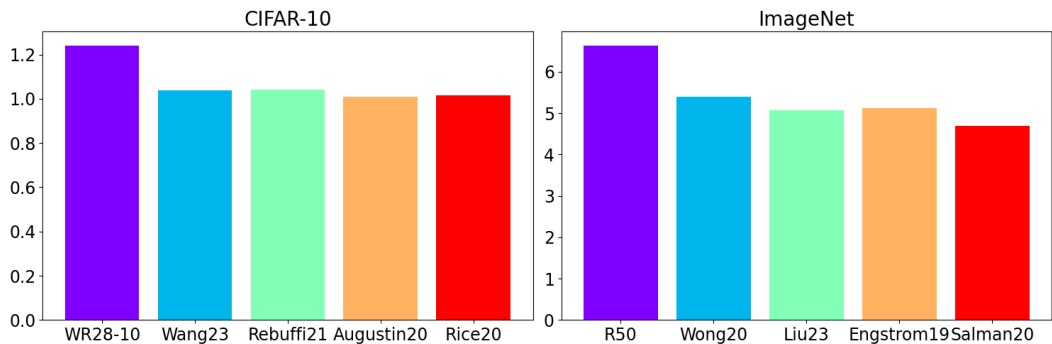

Figure 3: Nuclear norms of PGD attacks (10 steps) averaged over 5000 images from CIFAR-10 (left) and ImageNet (right) datasets.

Table 2: Comparison of robust accuracy $\rho$ across different datasets, models and algorithms. All attacks are computed with $\|\delta X\|_2 = 0.5$ for CIFAR-10 and $\|\delta X\|_2 = 0.25$ for ImageNet for classic PGD and $\|\delta U \otimes_C \delta V\|_{S^1} = \|\delta X\|_{S^1}$ for LoRa-PGD with 10 optimization steps. The best result per column is given in bold, and the second best is given in gray.

| | | CIFAR-10 | | | | ImageNet | | | | Rank |
| | Standard | Wa23 | Re21 | Au20 | Ri20 | Standard | Wo20 | Li23 | En19 | Sa20 | $r = \cdot\%R$ |
|---|---|---|---|---|---|---|---|---|---|---|---|
| **LoRa-PGD** $\|\cdot\|_{S^1}^{PGD} = \|\cdot\|_{S^1}^{LR}$ | 0.002 | **0.718** | **0.679** | **0.675** | **0.508** | **0.001** | **0.146** | **0.568** | **0.265** | **0.237** | 10% |
| | 0.002 | 0.776 | 0.738 | 0.726 | 0.588 | 0.001 | 0.181 | 0.604 | 0.3 | 0.278 | 20% |
| | 0.002 | 0.808 | 0.765 | 0.753 | 0.63 | 0.001 | 0.194 | 0.616 | 0.314 | 0.29 | 30% |
| | 0.002 | 0.816 | 0.774 | 0.764 | 0.643 | 0.001 | 0.203 | 0.621 | 0.318 | 0.297 | 40% |
| | **0.001** | 0.822 | 0.779 | 0.771 | 0.651 | 0.001 | 0.208 | 0.624 | 0.322 | 0.296 | 50% |
| Classic PGD | 0.018 | 0.85 | 0.801 | 0.813 | 0.698 | 0.004 | 0.304 | 0.675 | 0.425 | 0.399 | 100% |

significantly outperforms the full-rank PGD baseline across all models and datasets when the size of the perturbation is measured using the nuclear norm.

**Ablation Study on Initialization** The results shown above are obtained using a warm-up initialization strategy where a single step of full PGD is initially computed and the resulting perturbation is used to initialize the methods. To initialize the PGD attack, the initializing perturbation is taken as is, while for LoRa-PGD, we perform an SVD decomposition and thus obtain $U, V$ initializations by truncating the decomposition to the desired rank. To assess the extent of the impact that the initialization has on the performance of the approach, we report and compare in Table 2 the robust performance for two additional initialization strategies. Specifically,

- **Random Initialization.** This approach follows a similar procedure to LoRA Hu et al. (2022). Specifically, one of the matrices in the factorization is initialized from a Gaussian distribution, while the other is initialized with zeros.
- **Transfer Initialization.** This initialization tests the ability of LoRa-PGD to function as a transfer learner for adversarial attacks. To do this, we first compute a one-step full-rank PGD attack on the standard model and use it as the starting point for both full PGD and LoRa-PGD on the other models. For LoRa-PGD, we additionally compute the SVD on the attack to match the desired initialization rank.

We do not report the performance of full-rank PGD as its robust accuracy was essentially not affected by the choice of the initialization strategy.

### 4.2 RESULTS ON THE ADVERSARIAL TRAINING CAPABILITY

The results on adversarial performance in Table 1 for standard models indicate that LoRa-PGD can be particularly competitive in adversarial training when used to produce a more robust model against the specific choice of an attack. To show that, we conducted experiments to compare the effectiveness

Table 3: Comparison of robust accuracy $\rho$ across different datasets, models for LoRa-PGD. Each block corresponds to a different initialization strategy. All attacks are computed with $\|\delta X\|_2 = 0.5$ for CIFAR-10 and $\|\delta X\|_2 = 0.25$ for ImageNet wit 10 steps. For each section (each initialization) best result per column is highlighted.

| | | CIFAR-10 | | | | IMAGENET | | | | | RANK |
| | | STANDARD | WA23 | RE21 | AU20 | STANDARD | WO20 | LI23 | EN19 | SA20 | $r = \cdot\%R$ |
|---|---|---|---|---|---|---|---|---|---|---|---|
| RANDOM | **LoRa-PGD** | 0.08 | 0.886 | 0.836 | 0.847 | 0.752 | 0.017 | 0.355 | 0.696 | 0.472 | 0.455 | 10% |
| | | 0.042 | 0.871 | 0.821 | 0.831 | 0.729 | 0.01 | 0.349 | 0.692 | 0.464 | 0.445 | 20% |
| | | 0.025 | 0.864 | 0.816 | 0.824 | 0.719 | 0.007 | 0.344 | 0.689 | 0.461 | 0.442 | 30% |
| | | 0.02 | 0.862 | 0.813 | 0.822 | 0.716 | 0.006 | 0.342 | 0.689 | 0.462 | 0.441 | 40% |
| | | 0.017 | 0.861 | 0.813 | 0.821 | 0.717 | 0.006 | 0.342 | 0.69 | 0.46 | 0.438 | 50% |
| TRANSFER | **LoRa-PGD** | 0.07 | 0.887 | 0.836 | 0.846 | 0.75 | 0.014 | 0.356 | 0.695 | 0.469 | 0.449 | 10% |
| | | 0.03 | 0.869 | 0.82 | 0.829 | 0.726 | 0.006 | 0.342 | 0.69 | 0.459 | 0.438 | 20% |
| | | 0.015 | 0.861 | 0.814 | 0.821 | 0.716 | 0.004 | 0.338 | 0.687 | 0.454 | 0.433 | 30% |
| | | 0.013 | 0.859 | 0.812 | 0.818 | 0.715 | **0.003** | 0.336 | 0.687 | 0.454 | 0.432 | 40% |
| | | **0.009** | 0.858 | 0.81 | 0.817 | 0.714 | 0.003 | 0.335 | 0.686 | 0.453 | 0.43 | 50% |
| WARM-UP | **LoRa-PGD** | 0.07 | 0.883 | 0.833 | 0.844 | 0.747 | 0.014 | 0.347 | 0.692 | 0.46 | 0.44 | 10% |
| | | 0.031 | 0.867 | 0.819 | 0.828 | 0.724 | 0.007 | 0.336 | 0.688 | 0.452 | 0.432 | 20% |
| | | 0.015 | 0.862 | 0.812 | 0.82 | 0.713 | 0.004 | 0.33 | 0.688 | 0.447 | 0.427 | 30% |
| | | 0.013 | 0.858 | 0.81 | 0.817 | 0.712 | **0.003** | 0.329 | 0.686 | 0.444 | 0.427 | 40% |
| | | 0.01 | 0.857 | 0.809 | 0.816 | 0.711 | 0.003 | 0.326 | 0.686 | 0.446 | 0.426 | 50% |

Table 4: Comparison of adversarial training results for ResNet18 and WRN28-10 models and CIFAR-10 dataset. All attacks are computed with $\|\delta X\|_2 = 0.5$. Best results per column are highlighted.

| | Rank $r = \cdot\%R$ | ResNet18 | | | | | | | | WRN28-10 | | | | | | | |
| | | Random | | | | Warm-up | | | | Random | | | | Warm-up | | | |
| | | Clean$\rho$ | std | Robust$\rho$ | std | Clean$\rho$ | std | Robust$\rho$ | std | Clean$\rho$ | std | Robust$\rho$ | std | Clean$\rho$ | std | Robust$\rho$ | std |
|---|---|---|---|---|---|---|---|---|---|---|---|---|---|---|---|---|---|
| LoRa-PGD | 30% | **84.75** | 0.08 | 61.24 | 0.12 | 83.88 | 1.4 | 61.46 | 0.19 | 85.95 | 0.68 | 65.16 | 0.2 | **86.82** | 1.04 | 65.46 | 0.1 |
| | 40% | 82.57 | 1.89 | 60.78 | 0.83 | **84.53** | 0.26 | **61.68** | 0.13 | 86.47 | 1.3 | 65.82 | 0.52 | 85.71 | 1.38 | 65.64 | 0.67 |
| | 50% | 83.22 | 1.94 | **61.59** | 0.44 | 84.32 | 0.29 | **61.68** | 0.14 | **86.59** | 1.3 | **65.91** | 0.26 | 86.52 | 1.17 | **66.15** | 0.32 |
| | Classic PGD | 80.6 | 1.0 | 61.37 | 0.51 | 84.22 | 0.26 | 60.70 | 0.29 | 85.14 | 0.95 | 64.53 | 0.54 | 85.56 | 0.72 | 65.98 | 0.42 |

of adversarial training using LoRa-PGD against PGD given the same $L^2$ norm budget. We provide comparison on ResNet18 and WRN28-10 on CIFAR-10 in this subsection. We ran the experiment 5 times for each attack-initialization pairing and computed the average and standard deviation for standard and robust accuracy. Robust accuracy is tested against a PGD attack computed with 10 steps. For warm-up initialization, we use a pre-trained version of the corresponding model architecture that was provided in the MAIR framework and computed one-step PGD attack to be used as initialization for both PGD and LoRa-PGD. Remarkably, the results of these experiments show that adversarial training with LoRa-PGD produces a more robust model against PGD attack than PGD attack itself, despite using ranks that range from 30 to 40% of the maximal ones.

## 5 DISCUSSION AND LIMITATIONS

In this work, we observe that the classical PGD attack tends to generate perturbations that are numerically low-rank. Thus, we introduced LoRa-PGD, a simple low-rank variation of the PGD algorithm that efficiently generates adversarial attacks with a prescribed rank structure. At parity of run time and perturbation norm, LoRa-PGD outperforms standard PGD in terms of robust accuracy across a range of models and two image classification datasets. Furthermore, LoRa-PGD demonstrates superior performance in adversarial training, yielding improvements in both clean and robust accuracy. Finally, since LoRa-PGD shares some common features with the classical PGD attack, it also inherits certain limitations. In particular, gradient-based methods are generally ill-suited for black-box settings. Additionally, from a timing perspective, the computation of gradients imposes a lower bound on the potential speed improvements.

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

# A    NOTATION AND USEFUL DEFINITIONS

**Definition A.1.** ($\rho$bust accuracy) We define the robust accuracy $\rho$ through the use of the hard class function

$$C(\delta X) = \arg \max_{j=1,...,d} f_\theta^j(X + \delta X)$$

as follows:

$$\rho = \frac{1}{N_D} \sum_{i=1}^{N_D} \mathbb{1}_{C(0)}(C(\delta X_i))$$

**Definition A.2.** (Channelwise product, channelwise rank and channelwise norm)

Let $A \in \mathbb{R}^{C \times N \times K}, B \in \mathbb{R}^{C \times K \times M}$. We define the channelwise product of $A, B$, denoted by $A \otimes_C B \in \mathbb{R}^{C \times N \times M}$ as

$$(A \otimes_C B)_{cij} := \sum_{k=1}^{K} A_{cik} B_{ckj}$$

We define the channelwise rank of the tensor as:

$$\mathrm{rank}(A) := \max_{c=1,...,C} \mathrm{rank}(A_{c,:,:})$$

Moreover, for any matrix norm $\| \cdot \|_\omega : \mathbb{R}^{N \times K} \to \mathbb{R}_+$ denote with $\|A\|_\omega$ the channelwise norm, i.e.

$$\|A\|_\omega := \frac{1}{C} \sum_{c=1}^{C} \|A_{c,:,:}\|_\omega$$

Finally, we denote the transpose of a $A^\top$ the tensor $A^\top \in \mathbb{R}^{C \times K \times N}$ defined as:

$$A^\top_{cij} = A_{cji}$$

**Definition A.3.** (Gradient direction induced by a norm) Consider a function $\phi : U \subseteq H \to \mathbb{R}$ defined on a finite-dimensional real Hilbert space $H$ with inner product $h$, $U$ is an open subset of $H$, and let $\| \cdot \|$ be a norm on $H$. We say that a vector field $G : U \to H$ is the gradient of $\phi$ with respect to the norm $\| \cdot \|$ if:

$$G(x) = \begin{cases} \arg\max_{y \in H, \|y\| \leq 1} h(\nabla\phi(x), y), & \text{if } \nabla\phi(x) \neq 0 \\ 0, & \text{otherwise} \end{cases} \quad \forall x \in U$$

where $\nabla\phi(x)$ is the gradient induced by the inner product $h$, i.e. the only vector field that satisfies

$$h(\nabla\phi(x), v) = D\phi(x)[v], \quad \forall v \in H$$

We denote $G$ by $\widehat{\nabla}^{\|\cdot\|}\phi$ when there can be confusion, or if $\| \cdot \|$ is an $L^p$ norm, we will simply indicate $G = \widehat{\nabla}^{L^p}\phi$.

*Remark* A.4. (Existence and uniqueness of $\widehat{\nabla}^{\|\cdot\|}$) Since $H$ is finite dimensional, the ball $B_{\|\cdot\|}(0, 1]$ is compact (and convex), and the functional $y \mapsto h(\nabla\phi(x), y)$ is continuous and linear (therefore convex). A minimizer exists because of the Weierstrass theorem. Moreover, if $\nabla\phi(x) = 0$ then we have uniqueness, and if $\nabla\phi(x) \neq 0$, then $\widehat{\nabla}^{\|\cdot\|}\phi(x)$ is unique because $y \mapsto h(\nabla\phi(x), y)$ is strictly convex.

We also observe that the minimum value of the optimization problem is the dual norm

$$\min_{y \in H, \|y\| \leq 1} h(\nabla\phi(x), y) =: \|\nabla\phi\|_*.$$

Therefore, one even has a notion of magnitude for the gradient, which can be used in any optimization algorithm as scaling for the learning rate. Moreover, for $L^p$ norms, the dual is the usual $L^q$ with conjugate exponent.

In Appendix B, we will characterize entrywise-$L^p$ gradients of functions defined on matrix spaces $H = \mathbb{R}^{C \times N \times M}$ with the Frobenius inner product $h(X, Y) = \mathrm{Tr}(XY^\top)$.

# B  OPTIMAL $L^p$ ASCENT DIRECTIONS: $\widehat{\nabla}^{L^p}$ GRADIENT IN TENSOR SPACES

**Proposition B.1.** *(Optimal ascent direction in $L^p$ spaces). Consider the optimization problem Equation (1) where $\ell$ is substituted by its first order approximation around $\delta X = 0$, i.e.*

$$
\begin{cases}
\delta X^* \in \underset{\delta X \in \mathbb{R}^{C \times N \times M}}{\arg\max} \; \langle \nabla_X \ell(f_\theta(X), Y), \delta X \rangle \\
\|\delta X\|_{L^p} \leq \tau
\end{cases}, \tag{6}
$$

*where $\langle \cdot, \cdot \rangle$ represents the Frobenius inner product and $\nabla_X$ is the gradient with respect the Frobenius metric.*

*Then the solution of Equation (6) is given by*

$$
\delta X^* = \tau \frac{\mathrm{sign}(\nabla_X \ell(f_\theta(X), Y)) \odot |\nabla_X \ell(f_\theta(X), Y)|^{p-1}}{\||\nabla_X \ell|^{p-1}\|_{L^p}}
$$

*where $\odot$ represents the entrywise product and the power is entrywise.*

*Proof.* We use the Holder inequality, with $\frac{1}{p} + \frac{1}{q} = 1$:

$$
\langle \nabla_X \ell(f_\theta(X), Y), \delta X \rangle \leq |\langle \nabla_X \ell(f_\theta(X), Y), \delta X \rangle| \leq \sum_{c,i,j} |\partial_{X_{cij}} \ell (\delta X)_{cij}| \leq
$$

$$
\leq \sum_c \|\nabla_X \ell(f_\theta(X), Y)_{c,:,:}\|_{L^q} \|\delta X_{c,:,:}\|_{L^p}
$$

and equality is obtained for

$$
|\delta X_{cij}|^q = |\partial_{X_{cij}} \ell|^p, \quad \forall c, i, j
$$

which implies that the optimal perturbation has the form

$$
\delta X^*_{cij} = \tau \frac{\mathrm{sign}(\partial_{X_{cij}} \ell) |\partial_{X_{cij}} \ell|^{p/q}}{\||\nabla_X \ell|^{p/q}\|_{L^p}}
$$

By substituting the conjugate exponent $q = \frac{p}{p-1}$, we get the claimed result. $\qquad \square$

*Remark* B.2. (Common cases, $L^2$ and $L^\infty$ based attacks) We remark two particularly useful (and used) cases of Proposition B.1 are the ones in which $p = 2, p = \infty$. Given that the exponents are $p, q$ are conjugate, we have $q = \frac{p}{p-1}$. Therefore

$$
\delta X^* = \tau \frac{\mathrm{sign}(\nabla_X \ell(f_\theta(X), Y)) \odot |\nabla_X \ell(f_\theta(X), Y)|^{(p-1)}}{\||\nabla_X \ell|^{(p-1)}\|_{L^p}}
$$

which for $p = 2$ gives:

$$
\delta X^* = \tau \frac{\nabla_X \ell}{\|\nabla_X \ell\|_{L^2}},
$$

which is the classical $L^2$ based attack.

For $p = \infty$, we have instead:

$$
\lim_{p \to +\infty} \frac{|v|^{p-1}}{(\sum_i |v_i|^{p(p-1)})^{1/p}} = \begin{cases} 1, & \text{if } |v_i| = \max_j |v_j| \\ 0, & \text{otherwise} \end{cases}
$$

Therefore, for $p = \infty$, the optimal perturbation is:

$$
\delta X^* = \tau \, \mathrm{sign}(\nabla_X \ell(f_\theta(X), Y)),
$$

which is the classical FGSM attack Goodfellow et al. (2014).

Moreover, for $\tau = 1$, Proposition B.1 can be seen as a characterization of $L^p$ gradient. In fact, in view of Definition A.3, we have that

$$
\widehat{\nabla}_X^{L^p} \ell(f_\theta(X), Y) = \frac{\mathrm{sign}(\nabla_X \ell(f_\theta(X), Y)) \odot |\nabla_X \ell(f_\theta(X), Y)|^{(p-1)}}{\||\nabla_X \ell|^{(p-1)}\|_{L^p}}
$$

**Corollary B.3.** *(Optimal low-rank ascent direction). Consider the optimization problem Equation* (1) *where $\ell$ is substituted by its first order approximation around $\delta X = 0$, i.e.*

$$\begin{cases} \delta U^* \in \displaystyle\arg\max_{\delta U \in \mathbb{R}^{C \times N \times r}} \langle \nabla_X \ell(f_\theta(\bar{X}), Y), \delta U \otimes_C \delta V \rangle \\ \|\delta U\|_{L^p} \leq \tau \end{cases}, \tag{7}$$

*where $\langle \cdot, \cdot \rangle$ represents the Frobenius inner product and $\nabla_X$ is the gradient with respect the Frobenius metric.*

*Then the solution of Equation* (6) *is given by*

$$\delta U^* = \tau \frac{\operatorname{sign}(\nabla_X \ell(f_\theta(\bar{X}), Y) \otimes_C \delta V^\top) \odot |\nabla_X \ell(f_\theta(\bar{X}), Y) \otimes_C \delta V^\top|^{p-1}}{\||\nabla_X \ell \otimes_C \delta V^\top|^{p-1}\|_{L^p}}$$

*where $\odot$ represents the entrywise product and the power is entrywise.*

*Moreover, for $\bar{X} = X + \delta U \otimes_C \delta V$, we have by the chain rule that*

$$\delta U^* = \tau \frac{\operatorname{sign}(\nabla_{\delta U} \ell(f_\theta(\bar{X}), Y)) \odot |\nabla_{\delta U} \ell(f_\theta(\bar{X}), Y)|^{p-1}}{\||\nabla_{\delta U} \ell|^{p-1}\|_{L^p}}$$

*Proof.* Immediate from Proposition B.1 and the chain rule. □

*Remark B.4.* We remark that an analogous result holds for $\delta U$ fixed and maximizing on $\delta V$.

## C ADDITIONAL RESULTS

**Additional information on perturbations** In Table 5, we provide comparison of LoRa-PGD, Classic PGD and AutoAttack for 5 choices of attack magnitude for both CIFAR-10 and ImageNet with 5 models for each dataset. In Table 6, we show the changes of robust accuracy for Standard model for different amounts of steps taken by either LoRa-PGD or Classic PGD. To provide additional information on the time aspect we also provide Figure 6 and Figure 6 to show dynamic change of robust accuracy with time (average time for 100 images). We remark that we cannot compare our results with the ones given by Carlini-Wagner's attack Carlini & Wagner (2017), since the size of the maximal perturbation cannot be exactly enforced for every sample in the original formulation.

**Visual comparison** To further explore the differences between full-rank PGD and LoRa-PGD attacks, we present a visual comparison in Figure 4 using images from the CIFAR-10 dataset. Specifically, we compare the original unperturbed image with versions perturbed by PGD where the $L^2$ norm of the attack is fixed as $\|\delta X\|_{L^2} = 0.5$; LoRa-PGD with the same $L^2$ norm $\|\delta U \otimes_C \delta V\|_{L^2} = 0.5$, and for both rank=10%R and rank=50%R; LoRa-PGD with the nuclear norm adjusted to match that of the full PGD attack, i.e $\|\delta U \otimes_C \delta V\|_{S^1} = \|\delta X\|_{S^1}$. All methods employ 10 gradient steps on the CIFAR-10 dataset.

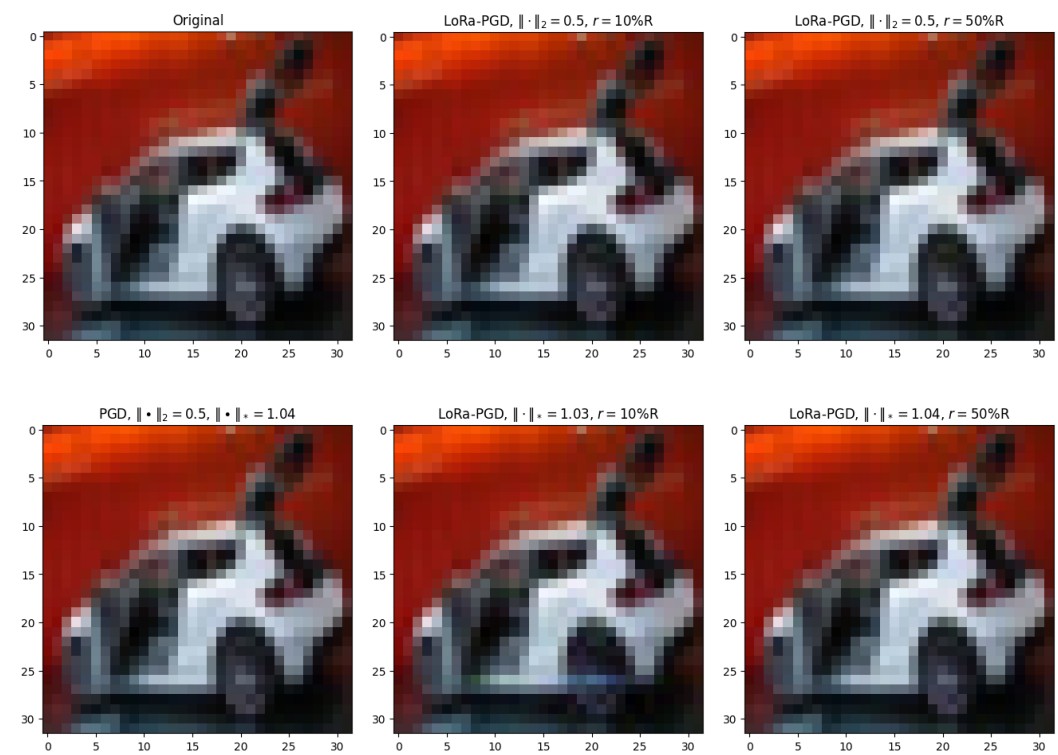

Figure 4: Perceivability examples, CIFAR-10 dataset, Wang23 model

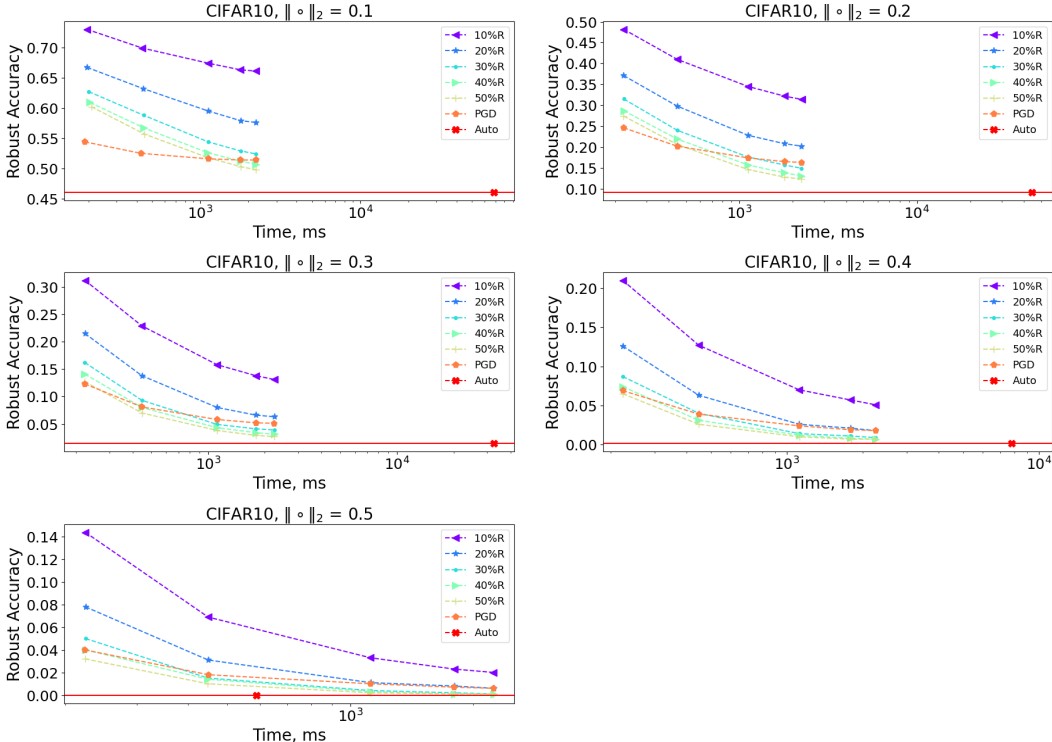

Figure 5: Comparison of different attacks' time vs robust accuracy for the Standard model on CIFAR-10.

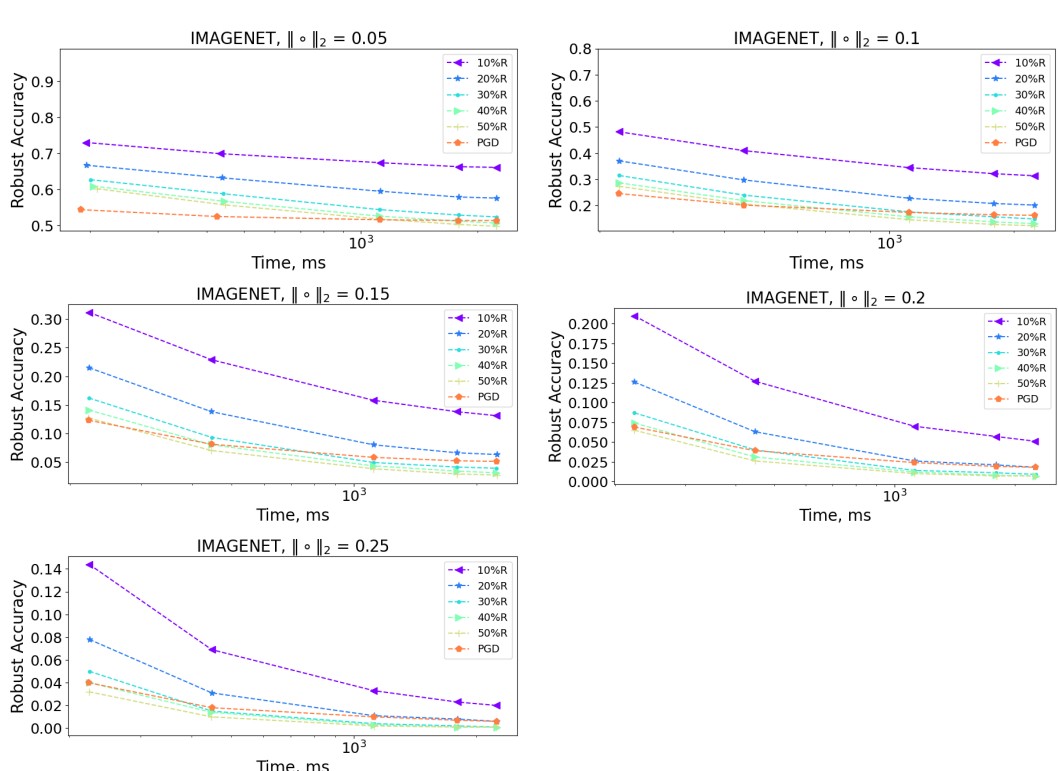

Figure 6: Comparison of different attacks' time vs robust accuracy for the Standard model on ImageNet.

Table 5: Comparison of robust accuracy $\rho$ across different datasets, models and algorithms, Steps=50

| | | CIFAR-10 | | | | | ImageNet | | | | | Ranks $r = \cdot\%R$ |
|---|---|---|---|---|---|---|---|---|---|---|---|---|
| | | St. | Wa23 | Re21 | Au20 | Ri20 | St. | Wo20 | Li23 | En19 | Sa20 | |
| **LoRa-PGD** $\|\cdot\|_2 = 0.1$ | | 0.662 | 0.941 | 0.9 | 0.926 | 0.862 | 0.334 | 0.757 | 0.489 | 0.599 | 0.598 | 10% |
| | $\|\cdot\|_2 = 0.05$ | 0.578 | 0.939 | 0.898 | 0.924 | 0.859 | 0.255 | 0.757 | 0.487 | 0.596 | 0.595 | 20% |
| | | 0.526 | 0.939 | 0.897 | 0.923 | 0.858 | 0.226 | 0.756 | 0.485 | 0.595 | 0.594 | 30% |
| | | 0.509 | 0.939 | 0.897 | 0.922 | 0.858 | 0.211 | 0.756 | 0.484 | 0.594 | 0.594 | 40% |
| | | 0.5 | 0.939 | 0.897 | 0.922 | 0.858 | 0.205 | 0.756 | 0.484 | 0.595 | 0.594 | 50% |
| Classic PGD | | 0.514 | 0.938 | 0.896 | 0.922 | 0.857 | 0.23 | 0.756 | 0.482 | 0.594 | 0.593 | 100% |
| Autoattack | | 0.461 | 0.937 | 0.894 | 0.921 | 0.855 | - | - | - | - | - | - |
| **LoRa-PGD** $\|\cdot\|_2 = 0.2$ | | 0.315 | 0.928 | 0.885 | 0.91 | 0.838 | 0.098 | 0.737 | 0.451 | 0.562 | 0.556 | 10% |
| | $\|\cdot\|_2 = 0.1$ | 0.202 | 0.924 | 0.883 | 0.904 | 0.832 | 0.052 | 0.735 | 0.443 | 0.559 | 0.55 | 20% |
| | | 0.15 | 0.922 | 0.88 | 0.901 | 0.828 | 0.04 | 0.735 | 0.441 | 0.557 | 0.549 | 30% |
| | | 0.134 | 0.922 | 0.88 | 0.9 | 0.826 | 0.035 | 0.734 | 0.44 | 0.557 | 0.548 | 40% |
| | | 0.126 | 0.922 | 0.88 | 0.9 | 0.826 | 0.032 | 0.734 | 0.439 | 0.557 | 0.547 | 50% |
| Classic PGD | | 0.163 | 0.921 | 0.879 | 0.9 | 0.825 | 0.048 | 0.733 | 0.436 | 0.556 | 0.546 | 100% |
| Autoattack | | 0.092 | 0.919 | 0.876 | 0.896 | 0.821 | - | - | - | - | - | - |
| **LoRa-PGD** $\|\cdot\|_2 = 0.3$ | | 0.134 | 0.913 | 0.871 | 0.889 | 0.812 | 0.029 | 0.718 | 0.404 | 0.521 | 0.506 | 10% |
| | $\|\cdot\|_2 = 0.15$ | 0.062 | 0.905 | 0.864 | 0.878 | 0.798 | 0.011 | 0.715 | 0.393 | 0.515 | 0.5 | 20% |
| | | 0.039 | 0.901 | 0.859 | 0.874 | 0.791 | 0.007 | 0.714 | 0.39 | 0.513 | 0.497 | 30% |
| | | 0.031 | 0.9 | 0.858 | 0.872 | 0.79 | 0.006 | 0.714 | 0.387 | 0.51 | 0.497 | 40% |
| | | 0.027 | 0.9 | 0.858 | 0.872 | 0.789 | 0.006 | 0.714 | 0.387 | 0.511 | 0.496 | 50% |
| Classic PGD | | 0.051 | 0.9 | 0.855 | 0.871 | 0.789 | 0.012 | 0.712 | 0.386 | 0.511 | 0.498 | 100% |
| Autoattack | | 0.015 | 0.896 | 0.848 | 0.861 | 0.78 | - | - | - | - | - | - |
| **LoRa-PGD** $\|\cdot\|_2 = 0.4$ | | 0.052 | 0.897 | 0.851 | 0.862 | 0.777 | 0.008 | 0.7 | 0.359 | 0.474 | 0.453 | 10% |
| | $\|\cdot\|_2 = 0.2$ | 0.019 | 0.888 | 0.839 | 0.849 | 0.755 | 0.003 | 0.697 | 0.346 | 0.465 | 0.444 | 20% |
| | | 0.01 | 0.882 | 0.831 | 0.844 | 0.746 | 0.003 | 0.696 | 0.34 | 0.463 | 0.441 | 30% |
| | | 0.007 | 0.88 | 0.829 | 0.841 | 0.744 | 0.002 | 0.696 | 0.338 | 0.461 | 0.443 | 40% |
| | | 0.006 | 0.879 | 0.829 | 0.841 | 0.743 | 0.002 | 0.696 | 0.337 | 0.461 | 0.442 | 50% |
| Classic PGD | | 0.018 | 0.878 | 0.827 | 0.842 | 0.74 | 0.004 | 0.695 | 0.343 | 0.467 | 0.448 | 100% |
| Autoattack | | 0.002 | 0.87 | 0.817 | 0.83 | 0.733 | - | - | - | - | - | - |
| **LoRa-PGD** $\|\cdot\|_2 = 0.5$ | | 0.021 | 0.879 | 0.827 | 0.835 | 0.739 | 0.003 | 0.681 | 0.32 | 0.428 | 0.405 | 10% |
| | $\|\cdot\|_2 = 0.25$ | 0.006 | 0.861 | 0.812 | 0.817 | 0.713 | 0.002 | 0.675 | 0.304 | 0.415 | 0.391 | 20% |
| | | 0.002 | 0.853 | 0.805 | 0.808 | 0.703 | 0.001 | 0.672 | 0.297 | 0.41 | 0.386 | 30% |
| | | 0.001 | 0.851 | 0.803 | 0.806 | 0.7 | 0.001 | 0.672 | 0.293 | 0.409 | 0.385 | 40% |
| | | 0.001 | 0.849 | 0.802 | 0.805 | 0.699 | 0.001 | 0.672 | 0.294 | 0.409 | 0.384 | 50% |
| Classic PGD | | 0.006 | 0.85 | 0.8 | 0.811 | 0.696 | 0.002 | 0.674 | 0.297 | 0.422 | 0.394 | 100% |
| Autoattack | | 0. | 0.835 | 0.785 | 0.786 | 0.681 | - | - | - | - | - | - |

Table 6: Comparison of robust accuracy $\rho$ across different step quantity and perturbation sizes, $\tau$ for Standard model.

| | $\tau =$ | CIFAR-10 | | | | | IMAGENET | | | | | RANKS $r = \cdot\%R$ |
|---|---|---|---|---|---|---|---|---|---|---|---|---|
| | | 0.1 | 0.2 | 0.3 | 0.4 | 0.5 | 0.05 | 0.1 | 0.15 | 0.2 | 0.25 | |
| STEPS = 5 | LoRA-PGD | 0.73 | 0.482 | 0.312 | 0.21 | 0.144 | 0.425 | 0.199 | 0.096 | 0.052 | 0.029 | 10% |
| | | 0.667 | 0.371 | 0.215 | 0.126 | 0.078 | 0.353 | 0.131 | 0.053 | 0.025 | 0.012 | 20% |
| | | 0.627 | 0.315 | 0.162 | 0.087 | 0.05 | 0.321 | 0.104 | 0.039 | 0.016 | 0.008 | 30% |
| | | 0.609 | 0.287 | 0.141 | 0.074 | 0.04 | 0.303 | 0.091 | 0.034 | 0.013 | 0.006 | 40% |
| | | 0.602 | 0.274 | 0.127 | 0.065 | 0.032 | 0.292 | 0.084 | 0.03 | 0.012 | 0.006 | 50% |
| | CLASSIC PGD | 0.544 | 0.246 | 0.123 | 0.069 | 0.04 | 0.265 | 0.072 | 0.023 | 0.011 | 0.006 | 100% |
| STEPS = 10 | LoRA-PGD | 0.699 | 0.41 | 0.229 | 0.127 | 0.069 | 0.391 | 0.154 | 0.061 | 0.028 | 0.013 | 10% |
| | | 0.632 | 0.298 | 0.138 | 0.063 | 0.031 | 0.315 | 0.096 | 0.032 | 0.012 | 0.006 | 20% |
| | | 0.588 | 0.24 | 0.093 | 0.04 | 0.015 | 0.285 | 0.075 | 0.023 | 0.009 | 0.004 | 30% |
| | | 0.567 | 0.219 | 0.08 | 0.031 | 0.014 | 0.269 | 0.066 | 0.018 | 0.006 | 0.003 | 40% |
| | | 0.557 | 0.206 | 0.07 | 0.026 | 0.01 | 0.259 | 0.06 | 0.016 | 0.006 | 0.003 | 50% |
| | CLASSIC PGD | 0.525 | 0.202 | 0.081 | 0.039 | 0.018 | 0.242 | 0.056 | 0.015 | 0.006 | 0.003 | 100% |
| STEPS = 25 | LoRA-PGD | 0.674 | 0.345 | 0.158 | 0.07 | 0.033 | 0.353 | 0.117 | 0.037 | 0.011 | 0.005 | 10% |
| | | 0.595 | 0.228 | 0.08 | 0.026 | 0.011 | 0.28 | 0.065 | 0.017 | 0.005 | 0.002 | 20% |
| | | 0.544 | 0.176 | 0.049 | 0.014 | 0.004 | 0.247 | 0.05 | 0.011 | 0.004 | 0.002 | 30% |
| | | 0.526 | 0.157 | 0.043 | 0.012 | 0.003 | 0.23 | 0.042 | 0.009 | 0.003 | 0.002 | 40% |
| | | 0.518 | 0.146 | 0.038 | 0.01 | 0.002 | 0.222 | 0.04 | 0.009 | 0.002 | 0.001 | 50% |
| | CLASSIC PGD | 0.516 | 0.174 | 0.058 | 0.024 | 0.01 | 0.232 | 0.049 | 0.013 | 0.005 | 0.002 | 100% |
| STEPS = 40 | LoRA-PGD | 0.663 | 0.322 | 0.138 | 0.057 | 0.023 | 0.339 | 0.102 | 0.031 | 0.008 | 0.003 | 10% |
| | | 0.579 | 0.208 | 0.066 | 0.021 | 0.008 | 0.262 | 0.055 | 0.012 | 0.004 | 0.002 | 20% |
| | | 0.529 | 0.157 | 0.041 | 0.011 | 0.002 | 0.232 | 0.043 | 0.008 | 0.003 | 0.001 | 30% |
| | | 0.512 | 0.138 | 0.034 | 0.008 | 0.001 | 0.216 | 0.037 | 0.007 | 0.002 | 0.001 | 40% |
| | | 0.503 | 0.128 | 0.029 | 0.007 | 0.001 | 0.211 | 0.034 | 0.006 | 0.002 | 0.001 | 50% |
| | CLASSIC PGD | 0.514 | 0.165 | 0.052 | 0.019 | 0.007 | 0.232 | 0.049 | 0.013 | 0.005 | 0.002 | 100% |
| STEPS = 50 | LoRA-PGD | 0.661 | 0.314 | 0.131 | 0.051 | 0.02 | 0.334 | 0.098 | 0.029 | 0.008 | 0.003 | 10% |
| | | 0.576 | 0.202 | 0.063 | 0.018 | 0.006 | 0.255 | 0.052 | 0.011 | 0.003 | 0.002 | 20% |
| | | 0.524 | 0.149 | 0.039 | 0.009 | 0.001 | 0.226 | 0.04 | 0.007 | 0.003 | 0.001 | 30% |
| | | 0.507 | 0.131 | 0.031 | 0.007 | 0.001 | 0.211 | 0.035 | 0.006 | 0.002 | 0.001 | 40% |
| | | 0.498 | 0.123 | 0.027 | 0.007 | 0.001 | 0.205 | 0.032 | 0.006 | 0.002 | 0.001 | 50% |
| | CLASSIC PGD | 0.514 | 0.163 | 0.051 | 0.018 | 0.006 | 0.23 | 0.048 | 0.012 | 0.004 | 0.002 | 100% |

