# OpenReview forum: "Low-Rank Adversarial PGD Attack"
_ICLR.cc/2026/Conference — ICLR 2026 Conference Withdrawn Submission_

### Official Review · Reviewer_GYab · 2025-10-19

**Soundness:** 3
**Presentation:** 3
**Contribution:** 2
**Rating:** 4
**Confidence:** 4

**Summary:**

The paper proposes LoRa-PGD, a low-rank parameterization of Projected Gradient Descent adversarial attacks motivated by the empirical observation that standard PGD perturbations are largely low rank in their singular-value spectra. Instead of optimizing a full-image perturbation, the method optimizes a compact factorization nuclear-norm budgets, yielding attacks that are as strong as full-rank PGD at similar compute cost and sometimes approaching AutoAttack effectiveness at far lower runtime. The authors also show that training models with LoRa-PGD adversaries improves robust accuracy compared to PGD-based adversarial training on CIFAR-10 and ImageNet subsets. Overall, the work turns a spectral insight into a simple, practical attack/training tool, though broader comparisons with recent attack and training baselines would better contextualize its gains.

**Strengths:**

- Clear, simple idea: turns an empirical low-rank observation about PGD perturbations into a practical factorized attack.
- Sensible theory lens: nuclear-norm budgeting and singular-spectrum analysis offer an intuitive explanation for why it works.
- Balanced evaluation across attack and defense, with transparent compute accounting that makes results easy to interpret.

**Weaknesses:**

The weaknesses are mostly about the baselines and reported results:
- Limited efficiency gains vs. PGD: Despite the claims, the performance/efficiency trade-off appears modest. Table 1 and especially Figure 2 suggest that when wall-clock time is accounted for, LoRa-PGD’s advantage over standard PGD is small. Tables 2 and 4 hint at gains in certain setups, but overall the time-normalized improvements look limited.

- Missing recent PGD-derived baselines: The paper omits several works that build on PGD to speed up adversarial training or improve robustness at similar cost. Prior methods (e.g., [1], [2] for training-time efficiency; [3], [4] for performance improvements without extra cost) should be cited and included as baselines to fairly contextualize LoRa-PGD’s benefits.

- Threat model narrowness (only 𝐿2): Results are restricted to L2 perturbations, whereas L_inf is more common in the literature and practice. The paper should report evaluations (attacks and training) to establish broader applicability and clarify whether the low-rank parameterization transfers across norms.

** I am willing to improve my score if the concern are adequately answers

[1] Adversarial Training for Free!
[2] A Simple Fine-tuning Is All You Need: Towards Robust Deep Learning Via Adversarial Fine-tuning
[3] Parametric Noise Injection: Trainable Randomness to Improve Deep Neural
Network Robustness against Adversarial Attack
[4] Learn2Perturb: an End-to-end Feature Perturbation Learning to Improve Adversarial Robustness

**Questions:**

- I am curious about other variants of this approach with different norms, or even the black box set up and transferability
- how could this setup be extended to more recent baselines and tasks, such as LLMs or VLMs (e.g. CLIP)

---

### Official Review · Reviewer_b3wV · 2025-10-28

**Soundness:** 3
**Presentation:** 3
**Contribution:** 3
**Rating:** 2
**Confidence:** 4

**Summary:**

This paper makes a key observation regarding the low-rank structure of adversarial perturbations and introduces LoRA-PGD, a novel low-rank approximation approach for iterative attack generation. The method achieves substantial efficiency gains over standard PGD without compromising attack success rates.

**Strengths:**

This paper identifies a common yet frequently overlooked characteristic of adversarial perturbations—their low-rank nature. This finding significantly reduces the computational time and resource requirements for generating adversarial examples, thereby enhancing the practical deployability of such attacks. The experimental validation is thorough and well-designed.

**Weaknesses:**

1. The insight regarding the low-rank nature of adversarial perturbations is valuable. However, the design of the proposed LoRa-PGD method appears relatively straightforward, as it essentially decomposes the adversarial perturbation into two matrices U and V for solution. Could the authors elaborate on other noteworthy aspects of the method's design?
2. While inference time is a consideration, the primary bottleneck in deploying adversarial attacks lies in their white-box dependency. The application of low-rank decomposition does not inherently address this fundamental limitation.
3. The runtime comparison presented in the line chart is not sufficiently clear, as it fails to effectively highlight the magnitude of speed improvement achieved by the proposed method.
4. The paper does not adequately demonstrate the computational resource savings achieved by LoRa-PGD.

**Questions:**

See weakness.

---

### Official Review · Reviewer_ERdb · 2025-10-29

**Soundness:** 3
**Presentation:** 3
**Contribution:** 2
**Rating:** 2
**Confidence:** 5

**Summary:**

This paper proposes LoRa-PGD, a low-rank variant of the standard PGD adversarial attack. The authors start from the empirical observation that PGD perturbations often exhibit approximate low-rank structure. Leveraging this, and inspired by the LoRa-style factorization, they reparameterize the perturbation as a product of two small matrices/tensors that enforce a channel-wise rank constraint r. The authors evaluate LoRa-PGD on CIFAR-10 and ImageNet using standard and robust models from RobustBench.

**Strengths:**

- The method is elegant and easy to understand. The authors impose a low-rank factorization in an elegant manner following the LoRa approach.
- The authors include several ablations to further understand the behavior of the proposed attack and showcase preliminary results on adapting the proposed attack for adversarial-training experiments.

**Weaknesses:**

### **Marginality of empirical improvements**
As the main concern of this paper, I found that most reported differences relative to PGD are extremely small (often in the third decimal place in Table 1). Differences like 0.826 –> 0.827 or 0.546 –> 0.547 are within typical experimental noise and should not be considered meaningful without statistical evidence. Indeed, related to this observation, there are no per-experiment standard deviations/confidence intervals, nor are there hypothesis tests that show whether the reported gains are significant and repeatable. In this regard, Table 4 reports the results of adversarial training with LoRa-PGD compared with standard PGD. However, the observed gains are marginal and largely overlap with the PGD results across most settings, suggesting that the proposed approach does not provide a substantially stronger improvement in practice. Furthermore, all these marginal improvements come at the cost of introducing an additional hyperparameter r (rank) in LoRa-PGD that must be tuned by the attacker before running the experiments.

The authors claim competitiveness or even superiority over AutoAttack; however, this claim is not convincingly supported. The experiments involving AutoAttack are only partially conducted, as results on ImageNet are omitted due to time and memory constraints. In my view, a claim of being comparable or better to AutoAttack requires broader and complete empirical verification. At the very least, AutoAttack should be evaluated on a reduced subset of ImageNet (e.g., 1k–5k images), since it represents a more challenging, high-resolution benchmark. Alternatively, a direct comparison with APGD or APGD-t could provide a fair, computationally feasible baseline. Without that, the claim is not properly empirically supported.


### **Incomplete baseline coverage**
A central limitation of the paper lies in the incompleteness of the baseline comparisons, which weakens the empirical foundation of the claimed improvements. In particular:
 - APGD is not included as an explicit baseline. APGD is a modern, stronger iterative attack and is often sufficient to characterize PGD-like baselines. AutoAttack includes APGD internally; omitting APGD but referencing AutoAttack is inconsistent.

 - PGD is run for only 50 iterations in many comparisons. Iterative attacks often require more iterations to converge [i, ii] (indicatively 100+); running PGD for only 50 steps biases the comparison toward faster-converging variants, which are not always optimal at convergence [ii]. What happens to the empirical gains of LoRa-PGD when enabling algorithms to converge?

 - The evaluation is pointwise (fixed epsilon) and does not include cumulative or budget-aware metrics found in modern benchmarks (e.g., AttackBench evaluations). In this regard, AttackBench [ii] is an established benchmark for fairly comparing attacks in shared environments under common assumptions, and it provides evaluations across a range of epsilon values.

[i] ​​ Indicators of Attack Failure: Debugging and Improving Optimization of Adversarial Examples, 2022

[ii] AttackBench Evaluating Gradient-based Attacks for Adversarial Examples, 2025


### **Motivations**
The paper shows empirically that PGD perturbations are low-rank (Figure 1). However, if standard PGD already produces low-rank perturbations, the central question becomes: why does explicitly constraining rank help? The paper does not convincingly explain the mechanism by which forcing low-rank structure yields faster or stronger attacks beyond the fact that it seems to match PGD’s implicit bias. In this regard, no theoretical analysis or toy example illustrates how a rank constraint changes optimization dynamics (e.g., reducing variance or focusing gradients). The proposed approach, in its current presentation, does not promise principled takeaways for future development.


### **Limited Scope**
All main experiments are in the L2 threat model. PGD is routinely applied under Linf and L1 settings as well, where the geometry is very different and the relationship between spatial rank and perturbation efficacy may not hold. The paper should either justify why the approach is inherently tied to L2, or provide experiments demonstrating that LoRa-PGD generalizes to L1 and Linf (or explicitly discuss failure cases).

**Minor comments**
- Figures use bright colors and are not colorblind-friendly.
- Figures 1 and 3 lack y-axis labels.



**Actionable points summary**
Revise experiments to include APGD baseline and an AutoAttack subset on ImageNet (or APGD at matched budgets). Add statistical summaries and sensitivity plots for r. Add at least one toy theoretical/empirical demonstration that explains why low-rank constraints help beyond PGD’s implicit bias. Improve figure accessibility and add clear runtime breakdowns. After these revisions, reassess whether the gains persist and are meaningful.

**Questions:**

The warm-up initialization uses an SVD of a single-step PGD, which implicitly uses full-rank PGD to seed LoRa-PGD. Does LoRA-PGD preserve the same runtime savings when using random initialization?

---

### Official Review · Reviewer_d3ui · 2025-10-31

**Soundness:** 3
**Presentation:** 2
**Contribution:** 3
**Rating:** 4
**Confidence:** 4

**Summary:**

The paper proposes LoRa-PGD, a variant of PGD-based adversarial training that introduces a low-rank projection on gradient updates to improve stability and reduce overfitting. The method aims to exploit the low-dimensional structure of adversarial perturbations, arguing that constraining updates within a learned low-rank subspace leads to more “generalizable” robustness. Experimental results show modest improvements in robust accuracy compared to standard PGD.

**Strengths:**

- The paper provides a clear and reproducible extension to adversarial training, with a reasonable motivation grounded in the empirical observation that adversarial perturbations often reside in low-dimensional manifolds.
- The implementation is straightforward and compatible with standard PGD training, which could make the approach potentially useful if its effects were better analyzed and justified.

**Weaknesses:**

1. The paper does not thoroughly compare with recent improvements to PGD-based adversarial training. Without these baselines, the claimed advantages of LoRa-PGD are difficult to evaluate in context.
2. The reported improvements are generally small (often within 0.2–0.4% robust accuracy) and in some cases within noise margins. It is unclear whether these differences are statistically significant or reproducible across random seeds.
3. The results show inconsistent trends as the rank varies—higher ranks sometimes reduce robustness rather than increase it—but no clear analysis or interpretation is provided.
4. The work stays at the algorithmic level without providing theoretical insight. As a result, the novelty feels incremental relative to existing PGD-based adversarial training literature.

**Questions:**

1. How does LoRa-PGD perform compared to more recent and stronger PGD variants under the same training setup?
2. Can the authors provide an explanation for the **non-monotonic trend** between the rank value and robust accuracy? Why do higher ranks sometimes degrade performance?
3. Are the observed improvements statistically significant? Results of mean and standard deviation over multiple random seeds may help.
4. Could the low-rank constraint affect convergence speed or computational efficiency?

---

### Note · Authors · 2025-11-15

I have read and agree with the venue's withdrawal policy on behalf of myself and my co-authors.